# Preliminary Results from the ACTyourCHANGE in Teens Protocol: A Randomized Controlled Trial Evaluating Acceptance and Commitment Therapy for Adolescents with Obesity

**DOI:** 10.3390/ijerph19095635

**Published:** 2022-05-05

**Authors:** Anna Guerrini Usubini, Roberto Cattivelli, Asia Radaelli, Michela Bottacchi, Giulia Landi, Eliana Tossani, Silvana Grandi, Gianluca Castelnuovo, Alessandro Sartorio

**Affiliations:** 1Istituto Auxologico Italiano IRCCS, Psychology Research Laboratory, 20145 Milan, Italy; u.guerrini@auxologico.it (A.G.U.); m.bottacchi@auxologico.it (M.B.); 2Department of Psychology, Catholic University of Milan, 20123 Milan, Italy; asia.radaelli01@icatt.it; 3Department of Psychology, University of Bologna, Viale Berti Pichat 5, 40127 Bologna, Italy; roberto.cattivelli@unibo.it (R.C.); giulia.landi7@unibo.it (G.L.); eliana.tossani2@unibo.it (E.T.); silvana.grandi@unibo.it (S.G.); 4Laboratory of Psychosomatics and Clinimetrics, Department of Psychology, University of Bologna, Viale Europa 115, 47023 Cesena, Italy; 5Istituto Auxologico Italiano, IRCCS, Experimental Laboratory for Auxo-Endocrinological Research, Piancavallo [VB], 28824 Verbania, Italy; sartorio@auxologico.it; 6Istituto Auxologico Italiano, IRCCS, Division of Auxology, Piancavallo [VB], 28824 Verbania, Italy

**Keywords:** childhood obesity, obesity rehabilitation, adolescents, acceptance and commitment therapy, psychological well-being, experiential avoidance and fusion

## Abstract

The study shows preliminary results of “The ACTyourCHANGE in Teens” project, a Randomized Controlled Trial aimed at evaluating the efficacy of an Acceptance and Commitment Therapy-based intervention combined with treatment as usual (ACT+TAU) compared to TAU only, for improving psychological well-being, psychological distress, experiential avoidance and fusion, emotion dysregulation, and emotional eating in a sample of 34 in-patient adolescents with obesity (Body Mass Index > 97th centile). Mixed between-within 2 × 2 repeated-measures analyses of variances (ANOVAs) were carried out to examine the changes in psychological conditions of participants over time. Moderation analyses were also conducted to test whether pre-test anxiety, depression, stress, and experiential avoidance and fusion predicted emotional eating at post-test with groups (ACT+TAU vs. TAU only) as moderators. Only a significant interaction effect (time × group) from pre- to post-test (*p* = 0.031) and a significant main effect of time on anxiety (*p* < 0.001) and emotional eating (*p* = 0.010) were found. Only in the TAU only group were higher levels of depression (*p* = 0.0011), stress (*p* = 0.0012), and experiential avoidance and fusion (*p* = 0.0282) at pre-test significantly associated with higher emotional eating at post-test. Although future replication and improvements of the study may allow us to obtain more consistent results, this preliminary evidence is actually promising.

## 1. Introduction

In recent years we have witnessed a rapid growth of childhood obesity rates that have reached alarming proportions, both in developed and developing countries [1,2]. As for adulthood, once established, obesity in children significantly influence their physical and psychological well-being. It has been found that obesity is strongly associated with some form of psychopathologies such as depression, anxiety, stress, low self-esteem, as well as social disadvantages such as bullying, peer victimization, and stigma [3,4,5]. In this scenario, females seem to be at a higher risk for developing depression and anxiety than males [6]. 

To manage obesity, multidisciplinary approaches entailing medical, nutritional, physical, and psychological components have been depicted as the most effective strategy to produce sustainable and significant effects on childhood obesity intervention [7]. From a psychological point of view, Cognitive Behavioral Therapy (CBT) is nowadays considered the gold standard treatment for childhood obesity. Psychological interventions inspired by CBT usually entail goal-setting, self-monitoring, psychoeducation, stimulus control, problem-solving, and cognitive restructuring [8]. Empirical evidence suggests that CBT leads to satisfactory results with a reduction between 8 to 10 % of the initial weight. However, weight loss maintenance over time remains challenging [9] and motivates research to find significant factors associated with long-standing disordered eating habits related to obesity. 

To address the challenge of weight management, the theoretical model of Forman and Butryn [9] proposed a self-regulation-based framework suggesting that several specific self-regulation skills are needed to maintain healthy eating and physical activity. These self-regulation skills include values, clarity, behavioral commitment, self-awareness of thoughts, feelings and bodily sensations, and distress tolerance. Those self-regulation skills were found to play a protective role against the onset of unhealthy eating habits [9], including emotional eating. For this reason, they may be considered a valuable target of interventions aimed to achieve long-term weight management. 

Self-regulation skills are the core target of Acceptance and Commitment Therapy (ACT). ACT is one of the well-established third wave CBTs [10] aimed to promote “psychological flexibility”, defined as the ability of “contacting the present moment fully as a conscious human being, and basing on what the situation affords, changing or persisting in behavior in the service of chosen values” [10]. The promotion of psychological flexibility is based on three pillars: *Openness, Awareness, and Engagement*. *Openness* entails the willingness to have an open attitude to accept unpleasant internal states such as feelings and thoughts even if they represent a source of suffering; *Awareness* refers to paying attention to personal thoughts and sensations, without automatically reacting; finally, *Engagement* refers to identifying and engaging in valued actions that one finds as chosen life directions. 

A substantial body of evidence supports that psychological flexibility is a protective factor associated with adaptive responses to distress and positive mental health outcomes across different contexts [11,12,13]. By allowing people to engage in meaningful challenges, while following personal self-concepts and important life domains, psychological flexibility could have the potential to promote well-being and increase the adherence to a healthy lifestyle. To date, we have found only one pilot study assessing the efficacy of a 16-week, group ACT-based lifestyle modification treatment for adolescents and their parents/guardians that showed promising results in terms of reduced BMI (−1.3%), increased cognitive restraint, reduced hunger, and increased physical activity. In addition, results showed improvements in most quality of life domains and depression. Although results are promising, additional findings are needed. 

The present work aims to present the preliminary results of “The ACTyourCHANGE in Teens” study [14] a Randomized Controlled Trial (RCT) evaluating the efficacy of a brief ACT-based psychological intervention combined with treatment as usual (TAU) compared to TAU only in improving psychological conditions in a sample of adolescents with obesity within the context of a wider in-hospital multidisciplinary rehabilitation program for weight loss. This work includes preliminary results of the efficacy of our intervention in improving psychological conditions of adolescents with obesity. In this regard, our hypothesis was to find significant improvements in psychological well-being, a reduction of psychological distress, emotion dysregulation, experiential avoidance and fusion, and emotional eating in the ACT+TAU group compared to the TAU only group. 

The second aim of this study was to explore potential mechanisms implicated in the therapeutic benefits in emotional eating of such an intervention. In particular, we examined whether initial levels of psychological distress (i.e., anxiety, depression, and stress), emotion dysregulation, and experiential avoidance and fusion had an influence on levels of emotional eating after the intervention, considering the two groups of intervention by addressing moderation hypotheses. Those results were obtained by the first wave of data collection of the project. 

## 2. Method

### 2.1. Participants and Procedures

The sample consisted of 34 obese adolescents, 7 males, 27 females, aged between 13 and 17 years, with an average BMI of 38.2. Participants were recruited at the Division of Auxology Istituto Auxologico Italiano IRCCS, Piancavallo (VB), located in Northwest Italy, a specialized clinical center (i.e., third level) offering a 3-week in-hospital body weight reduction program. Participants were selected for the study if they were aged between 12 and 17, had a BMI > 97th centile according to age- and sex-specific Italian charts [15], and if they were of Italian mother tongue. Exclusion criteria were the presence of any physical or psychiatric problems according to the Diagnostic and Statistical Manual of Mental Disorders (DSM-5) criteria that could compromise participation in the study. 

Participants were selected at the admission to the hospital (Time 0) and screened for participating in the study with a clinical interview conducted by a clinical psychologist blinded to research aims, to provide information about the study and assess the presence of the abovementioned exclusion criteria. Weight and height, to calculate BMI (kg/m^2^), were assessed by the medical team. After obtaining informed consent from parents and written assent from participants, youth were asked to complete a battery of self-report questionnaires to collect demographical and clinical variables of interest at pre-test. The questionnaires were filled out under the supervision of a member of the research team. After completing the pre-test assessment at Time 0 (week 1), participants were randomly assigned into the following two conditions:−ACT+TAU group: Participants assigned to this group attended the standard 3-week multidisciplinary rehabilitation program plus a brief ACT-based intervention;−TAU only group: Patients received the standard 3-week multidisciplinary rehabilitation program only.

At the end of the 3-week residential rehabilitation program, just before discharge, all participants were asked to complete the same post-test assessment at Time 1 (week 3). From here on, we referred to pre-test/ Time 0/ week 1 as pre-test and to post-test/ Time 1/ week 3 as post-test.

The pre-test assessment was completed before the randomization, in order to ensure similar baseline characteristics of groups. Randomization with 1:1 allocation ratio was performed using the Web site Randomization.com [http://www.randomization.com].

The published study protocol of the study was registered on ClinicalTrials.gov (ID: NCT04896372) and approved by the Ethical Committee of the Istituto Auxologico Italiano (approval number: 2021_01_26_03). All procedures were conducted following the Helsinki Declaration and its later advancements. Procedures of the study were scheduled in Figure 1.

### 2.2. Measures

Demographical (gender, age, nationality, educational level, and family composition) and clinical data (psychological well-being, psychological distress, experiential avoidance and fusion, emotion dysregulation, and emotional eating) were collected via self-report using Italian validated and widely used questionnaires. Weight and height were measured at admission to the hospital by the medical team. BMI was calculated using the following formula: Kg/m^2^.

The primary outcome of the study was psychological well-being. The *Psychological Well-Being Scales* (PWB) [16]; Italian version [17] is a self-report measure used to assess psychological well-being that explores six dimensions: self-acceptance, positive relationships with others, autonomy, environmental control, personal growth, and life purpose. The questionnaire consists of 18 items rated on a 4-point Likert scale ranging from 1 (completely disagree) to 4 (completely agree). In our sample, the Cronbach’s alpha of the total score was 0.70.

Secondary outcomes included psychological distress, emotion dysregulation, experiential avoidance and fusion, and emotional eating.

The *Depression Anxiety Stress Scale* (DASS-21) ([18]; Italian version [19]) is a widely used [20,21] measure of psychological distress. It consists of 21 items rated on a 4-point Likert scale, ranging from 0 to 3 composing three subscales: depression (DASS-D), anxiety (DASS-A), and stress (DASS-S). In our sample, the Cronbach’s alpha of depression, anxiety, and stress subscales were, respectively, 0.91, 0.88, and 0.86. 

The *Difficulties in Emotion Regulation Scale* (DERS) ([20]; Italian version [21]) was administered to assess difficulties in emotional dysregulation. This is a self-report questionnaire consisting of 36 items, rated on a 5-point Likert scale ranging from 1 (almost never) to 5 (almost always), which explores the following subscales: non-acceptance of negative emotions, inability to undertake purposeful behavior when experiencing negative emotions, difficulty in controlling impulsive behavior when experiencing negative emotions, limited access to emotion regulation strategies that are considered effective, lack of awareness of one’s emotions, lack of understanding of the nature of one’s emotional responses. The Italian version has been widely used in samples of adolescents [22,23]. In our sample, the Cronbach’s alpha of the total score was 0.94. 

The *Avoidance and Fusion Questionnaire for Youth* (AFQ-Y) ([24]; Italian version [25]) was used as a measure of experiential avoidance and fusion in adolescents. It consists of 8 items rated on a 5-point Likert scale ranging from 0 (not at all true) to 4 (absolutely true). In our sample, the Cronbach’s alpha was 0.90.

The Emotional Eating subscale of the *Dutch Eating Behavior Questionnaire* (DEBQ) ([26]; Italian version [27]) was used to assess emotional eating (DEBQ-EE). The DEBQ is a self-report questionnaire used to detect eating behaviors. The Emotional Eating subscale consists of 13 items, rated on a 5-step Likert scale ranging from 0 (never) to 4 (almost always). In our sample, the Cronbach’s alpha of the total score was 0.97. 

### 2.3. Intervention

#### 2.3.1. The Multidisciplinary Rehabilitation Program for Weight Loss (TAU)

The hospital where the study was conducted offers a multidisciplinary inpatient treatment program for weight management with medical, dietetical, physical, and psychological components that was followed by all the participants at the study according to the Italian National Health System recommendations [28,29,30,31]. The nutritional component of the intervention comprised a nutritional assessment carried out by the nutritionist staff, an individualized hypocaloric diet entailing an energy intake about 500 kcal lower than the resting energy expenditure, composed of 53% carbohydrates, 26% fat, and 21% protein [32], and a fluid intake of at least 1500 mL per day-1, and a daily nutritional counselling program comprising dietetics lessons. The physical rehabilitation was composed of a physical activity program consisting of two 30-minute sessions per day of cycling, walking, stationary rowing, and stretching under the supervision of physical trainers and medical monitoring. The psychological component of the intervention comprised weekly individual psychological counselling sessions, lasting about one hour each, aimed at promoting a healthy lifestyle and addressing psychological factors related to the onset of dysfunctional lifestyle habits.

#### 2.3.2. The Acceptance and Commitment Therapy-Based Intervention (ACT)

The proposed intervention was designed by the authors of the study basing on previous ACT-based weight management interventions [33,34] and adapted for this specific population. It was developed following the main ACT-based manuals [33,34], with adjustments according to the users [35,36] and the context of the study implementation. As detailed in the published study protocol, the intervention was aimed at promoting the three pillars of the Psychological Flexibility model [34] namely Openness, Awareness, and Engagement using theoretically rooted and manualized experiential exercises and key metaphors in the field of ACT. The intervention was composed of three sessions, provided once a week lasting about one hour each. The sessions were carried out by a licensed clinical psychologist with proven expertise in ACT clinical practice for adolescents both in individual and group settings, blinded to research aims.

### 2.4. Statistical Analysis

Descriptive statistics were conducted to explore the demographic and baseline profile of the sample and to check if the study variables were normally distributed. Frequencies and percentages were computed for categorical variables, means, and standard deviations for continuous variables. A series of independent samples t-tests were also conducted to examine whether the two treatment groups (ACT+TAU vs. TAU only) were different in any demographical and clinical variables at pre-test.

To compare groups, mixed between-within 2 (groups: ACT+TAU vs. TAU only) × 2 (times: pre-test vs. post-test) repeated measures analyses of variances (ANOVAs) were conducted to examine changes in means of scores of PWB, subscales of DASS-21, DERS, AFQ-Y, and DEBQ (Emotional Eating subscale) between groups over two measurement timepoints. Homogeneity of variances were tested using Levene tests. Effect size (η^2^) was used to quantify the global difference of the two groups across times. Effects size were interpreted with the following benchmarks [37]: null (η^2^ < 0.003); small (0.003 < η^2^ > 0.003 to 0.039); moderate (0.110 < η^2^ > 0.40); and large (η^2^ > 0.110).

As part of exploratory analysis, a series of moderation analyses were carried out. Specifically, we first tested the presence of a significant interaction between pre-test psychological distress (i.e., anxiety, depression, and stress) and the treatment groups (ACT+TAU vs. TAU only) in predicting emotional eating at post-test. Secondly, we examined the presence of a significant interaction between experiential avoidance and cognitive fusion at pre-test and the treatment groups (ACT+TAU vs. TAU only) in predicting emotional eating at post-test. Psychological distress and experiential avoidance and cognitive fusion were mean centered in order to reduce potential problems of multicollinearity and improve the interpretation of the coefficient in the interaction [38]. In the first set of moderations, we controlled for changes in experiential avoidance and cognitive fusion from pre-test to post-test, while in the second moderation we controlled for experiential avoidance and cognitive fusion at post-test. To provide a visual summary of moderations, values of pre-test psychological distress, and experiential avoidance and cognitive fusion (i.e., high = one standard deviation above the mean, average = mean, and low = one standard deviation below the mean) were selected and the conditional effect of the treatment groups at those values of pre-test psychological distress and experiential avoidance and cognitive fusion were estimated, and a final plot was created.

Participants who reported missing data or did not complete questionnaires or who dropped out from the program were excluded and their data were not analyzed (See Figure 2). Analyses were performed using Jamovi (The jamovi project 2021). jamovi (Version 1.6) [Computer Software]. Retrieved from https://www.jamovi.org.

## 3. Results

### 3.1. Baseline Descriptive of the Sample

Of the 34 adolescents recruited for the study, 17 were assigned to the ACT+TAU group and 17 were assigned to the TAU only group. In the ACT+TAU group 13 were females, while four were males. The mean of age was 15.5 (*SD* = 1.37) and the mean of BMI at pre-test was 38.5 (*SD* = 6.00). In the TAU only group, 14 were females, while three were males. The mean of age was 15.6 (*SD* = 1.06) and the mean of BMI at pre-test was 36.8 (*SD* = 6.47).

A series of independent samples t-tests revealed that at pre-test there were no significant difference between the groups (ACT+TAU vs TAU only) in psychological well-being (PWB) depression (DASS-D), anxiety (DASS-A), stress (DASS-S), emotion dysregulation (DERS), experiential avoidance and fusion (AFQ-Y), and emotional eating (DEBQ-EE). Results are shown in Table 1.

### 3.2. Preliminary Evidence of Treatment Efficacy

A series of mixed 2 × 2 (between-within) repeated measures ANOVAs were performed to analyze differences between the two groups of interventions (ACT+TAU vs TAU only) at two measurement timepoints (pre-test vs. post-test) in psychological well-being (PWB), psychological distress, specifically depression (DASS-S), anxiety (DASS-A) and stress (DASS-S), emotional dysregulation (DERS), experiential avoidance and fusion (AFQ-Y), and emotional eating (DEBQ-EE).

As for psychological well-being, results showed no significant interaction effect of time x group on PWB from pre-test to post-test (*F*(1,29) = 0.000159; *p* = 0.990; ƞ^2^ = 0.000) and no significant main effect of the within factor of time (*F*(1,29) = 0.152; *p* = 0.699; ƞ^2^ = 0.002). The main effect of the between factor of group was not significant (*F*(1,29) = 0.00314; *p* = 0.0956; ƞ^2^ = 0.000).

As for depression, the results showed no significant interaction effect of time x group on DASS-D from pre-test to post-test (*F*(1,32) = 0.0762; *p* = 0.784; ƞ^2^ = 0.000) and no significant main effect of the within factor of time (*F*(1,32) = 1.4943; *p* = 0.230; ƞ^2^ = 0.006). The main effect of the between factor of group was not significant (*F*(1,32) = 0.102; *p* = 0.752; ƞ^2^ = 0.003).

As for anxiety, results showed a significant interaction effect of time x group on DASS-A from pre-test to post-test (*F*(1,32) = 5.11; *p* = 0.031; ƞ^2^ = 0.017) and a significant main effect of the within factor of time (*F*(1,32) = 159.00; *p* < 0.001; ƞ^2^ = 0.534). The main effect of the between factor of group was not significant (*F*(1,32) = 1.06; *p* = 0.312; ƞ^2^ = 0.011).

As for stress, results also showed no significant interaction effect of time x group on DASS-S from pre-test/Time 0/week 1 to post-test/Time 1/week 3 (*F*(1,32) = 0.0395; *p* = 0.844; ƞ^2^ = 0.000) and no significant main effect of the within factor of time (*F*(1,32) = 2.3192; *p* = 0.138; ƞ^2^ = 0.016). The main effect of the between factor of group was not significant (*F*(1,32) = 1.16 × 10^−31^; *p* = 1.000; ƞ^2^ = 0.000).

As for emotion dysregulation, results showed no significant interaction effect of time x group on DERS from pre-test to post-test (*F*(1,32) = 0.971; *p* = 0.332; ƞ^2^ = 0.001) and no significant main effect of the within factor of time (*F*(1,32) = 2.350; *p* = 0.135; ƞ^2^ = 0.003). The main effect of the between factor of group was not significant (*F*(1,32) = 1.88; *p* = 0.180; ƞ^2^ = 0.053).

As for experiential avoidance and fusion, results showed no significant interaction effect of time x group on AFQ-Y from pre-test to post-test (*F*(1,32) = 1.11; *p* = 0.299; ƞ^2^ = 0.003) and no significant main effect of the within factor of time (*F*(1,32) = 3.61; *p* = 0.067; ƞ^2^ = 0.011). The main effect of the between factor of group was not significant (*F*(1,32) = 0.385; *p* = 0.539; ƞ^2^ = 0.011).

As for emotional eating, results showed no significant interaction effect of time x group on AFQ-Y from pre-test to post-test (*F*(1,32) = 0.0173; *p* = 0.896; ƞ^2^ = 0.000) but a significant main effect of the within factor of time (*F*(1,32) = 7.3942; *p* = 0.010; ƞ^2^ = 0.011). The main effect of the between factor of group was not significant (*F*(1,32) = 0.0865; *p* = 0.771; ƞ^2^ = 0.003).

Means scores are summarized in Table 2.

Results were subjected to an a posteriori power analysis for repeated measures with a mixed within-between interaction. Results revealed that given a sample size of 34, an α of 0.05, and a small effect size obtained (0.2), the achieved power (1-β) was 0.61.

### 3.3. Moderation Analyses

To address our moderation hypotheses, a series of moderation analyses were run. The first set of moderations assessed the influence of depression (DASS-D), anxiety (DASS-A), and stress (DASS-S) at pre-test as predictors, and on emotional eating (DEBQ-EE) at post-test as an outcome, considering the treatment group (ACT+TAU vs. TAU only) as a moderator.

As for depression, results showed a significant interaction between the pre-test and the two treatment groups (ACT+TAU vs TAU only) in predicting post-test DEBQ-EE (unstandardized coefficient = −0.0876, SE = 0.0348, *p* = 0.0175, ΔR^2^ = 0.1493). Among those in the TAU only group, higher levels of DEBQ-D at pre-test were significantly associated with a higher DEBQ-EE at post-test (unstandardized coefficient = 0.0893, SE = 0.0245, *p* = 0.0011). Among those in the ACT+TAU group, there was no significant relationship between pre-test DASS-D and the subsequent DEBQ-EE at post-test (unstandardized coefficient = 0.0017, SE = 0.0251, *p* = 0.9479).

As for anxiety, the results showed no significant interaction between pre-test DASS-D and the two treatment groups (ACT+TAU vs TAU only) in predicting post-test DEBQ-EE (unstandardized coefficient = −0.0214, SE = 0.0347, *p* = 0.5425, ΔR^2^ = 0.0077).

As for stress, results showed a significant interaction between pre-test DASS-S and the two treatment groups (ACT+TAU vs TAU only) in predicting post-test DEBQ-EE (unstandardized coefficient = −0.0776, SE = 0.0367, *p* = 0.0430, ΔR^2^ = 0.1040). Among those in the TAU only group, higher levels of DEBQ-S at pre-test were significantly associated with a higher DEBQ-EE at post-test (unstandardized coefficient = 0.0991, SE = 0.0275, *p* = 0.0012). Among those in the ACT+TAU group, there was no significant relationship between pre-test DASS-S and subsequent DEBQ-EE at post-test (unstandardized coefficient = 0.0215, SE = 0.0242, *p* = 0.3809).

Additionally, we ran a moderation assessing the influence of emotion dysregulation (DERS) at pre-test in predicting emotional eating (DEBQ-EE) at post-test, considering the group as a moderator (ACT+TAU vs. TAU only). The results showed no significant interaction between pre-test DERS and the two treatment groups (ACT+TAU vs TAU only) in predicting post-test DEBQ-EE (unstandardized coefficient = −0.0030, SE = 0.0110, *p* = 0.7850, ΔR^2^ = 0.0012).

Finally, a moderation analysis was conducted to assess the impact of experiential avoidance and fusion (AFQ-Y) at pre-test in predicting emotional eating (DEBQ-EE) at post-test considering the group as a moderator (ACT+TAU vs TAU only). The results showed a significant interaction between pre-test AFQ-Y and the two treatment groups (ACT+TAU vs TAU only) in predicting post-test DEBQ-EE (unstandardized coefficient = −0.0417, SE = 0.0199, *p* = 0.0443, ΔR^2^ = 0.0626). As predicted, among those in the TAU only group, higher levels of AFQ-Y at pre-test were significantly associated with higher DEBQ-EE at post-test (unstandardized coefficient = 0.0437, SE = 0.0189, *p* = 0.0282). In contrast, among those in the ACT+TAU group, there was no significant relationship between pre-test AFQ-Y and subsequent DEBQ-EE at post-test (unstandardized coefficient = 0.0019, SE = 0.0195, *p* = 0.9224. The results are depicted in Figure 3.

## 4. Discussion

The main purpose of this study was to assess preliminary evidence of the efficacy of an ACT-based intervention added to a standard multidisciplinary rehabilitation program for weight loss in a group of adolescents with obesity to improve their psychological conditions.

The preliminary, but well-promising results showed that participants on both interventions (ACT+TAU vs. TAU only) reduced emotional eating. This result was in line with our expectations, even if it is not directly attributable to our intervention, since the reduction of emotional eating we observed was independent of the intervention received. In fact, results showed significant changes in emotional eating over time from pre-to-post intervention, independently from the intervention. However, the reduction in emotional eating is supposed to be a promising achievement for psychological interventions for childhood obesity, since emotional eating was found to be a significant well-known risk factor for weight gain in children [1].

As for other outcomes, we did not find significant changes except for anxiety. In fact, we found that levels of anxiety differently and significantly increased over time in the two groups, with a higher increase in anxiety in the ACT+TAU group than in the TAU only group over time. Even if this finding was unexpected, it is reasonably attributable to many reasons that, however, require further investigations. For example, since the DASS-21—the questionnaire we used to assess anxiety—asked participants to refer to the last week in their responses, it is possible to hypothesize that results reflected an effect of hospitalization (especially in the post-test), that it was found to be a stressful experience able to increase anxiety in pediatric age [39]. Another ACT-consistent hypothesis, and so, more suitable to explain the specific effect of the intervention, is that ACT promotes a change in relationships between individuals and their private events, such as thoughts, feelings, and sensations, and not the events themselves. Specifically, by promoting psychological flexibility, ACT suggests that the focus of the intervention should be on how the individual interacts with these thoughts, rather than reducing their form or frequency [40]. In light of this perspective, an increase in anxiety could reflect improved awareness about personal thoughts and emotional experiences and not necessarily, an increase in suffering. Consistent with this hypothesis, in our sample we found that levels of emotional eating—that generally are positively related to anxiety or stress [41]—decreased.

The second aim of the study, given its preliminary nature, was to explore associations between the variables of interest to speculate about the mechanism of action of our intervention. Moderation analyses revealed some interesting—however preliminary—findings. In fact, only in those who attended the TAU only intervention with higher levels of depression, stress, and experiential avoidance and fusion before the interventions were significant in predicting higher levels of emotional eating after the intervention, even if emotional eating decreased from pre-to-post intervention. The same relations did not emerge in those who received the ACT+TAU intervention, in which the link between initial psychological distress (specifically depression and stress) and emotional eating at the end of the intervention is not maintained. Again, such results may reflect the action of improved psychological flexibility that is considered a fundamental aspect of health [11] able to produce long-standing behavioral change, as in the case of healthy eating behaviors by increasing commitment to value-driven behaviors, by strengthening a willing, open, and accepting attitude of experiencing psychological events, and by improving the awareness of one’s internal and external experiences.

Our results need to be viewed in light of some limitations. In this study, the major limitation is represented by the relatively small sample size that may prevent the findings from being extrapolated, and warrants caution in drawing and interpreting conclusions. Another limitation concerns the lack of follow-up measures that may help to define the long-term effect of the intervention.

In addition, the high specificity of the context (third level Obesity clinic) where the study took place, limits the interpretations of results and their generalizability. However, the highly controlled environment where participants lived for 3 weeks—during research—allowed us to control for possible confounding factors related to childhood obesity (i.e., food intake, physical activity, family factors) [1]. A further limitation concerns the battery of self-report instruments used in the present study, which could be limited in measuring the multidimensional core process of psychological flexibility and could be affected by biases. In particular, it could be helpful to use the Multidimensional Psychological Flexibility Inventory to assess global psychological flexibility and each sub-component: acceptance, defusion, present moment awareness, self as a context, values, and committed actions [42], after being validated in adolescents. In addition, research about emotional eating suggests that using direct instead of retrospective measures to assess emotional eating and implementing innovative methodological tools such as the Ecological Momentary Assessment [43], could be free from the biases of self-report forms.

## 5. Conclusions

Even if preliminary, our results suggest that ACT could be a promising approach to be implemented in a context of an obesity rehabilitation program for adolescents, supporting the suitability of ACT in healthcare settings. However, the research program remains open, since further results need to be collected. Future directions of the study are oriented to achieve a larger sample size and collect data over time, adding follow-up measures. Doing so, it could also be interesting to assess whether any improvements in psychological flexibility would have a potentially positive impact on weight management over time, since preliminary evidence of reductions in emotional eating was found. Data collection of weight, BMI, and other physical measures at pre-test, post-test, and at follow-up will be implemented. In addition, we are also planning to compare the experimental conditions with a control group in order to better understand the impact of the rehabilitation program on treatment outcomes and to add other specific measurements for emotional eating.

## Figures and Tables

**Figure 1 ijerph-19-05635-f001:**
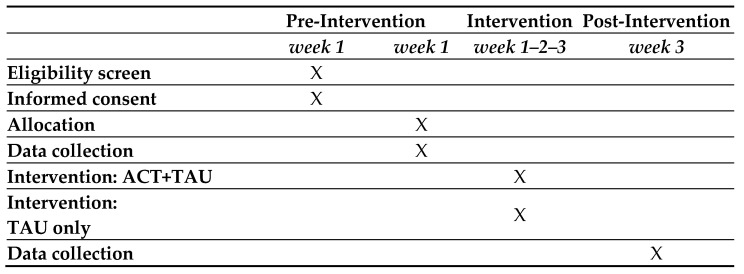
Schedule of enrollment, assessment, and intervention.

**Figure 2 ijerph-19-05635-f002:**
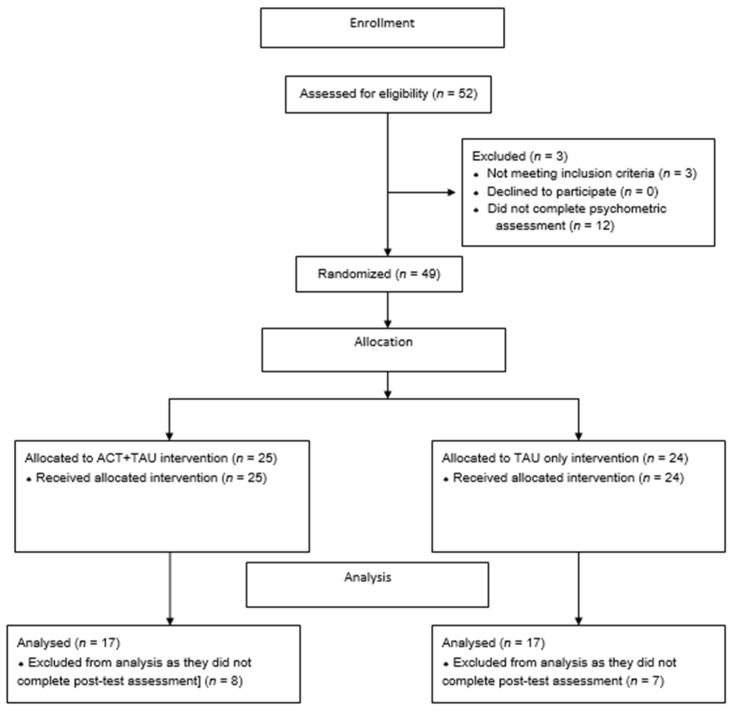
Flow chart.

**Figure 3 ijerph-19-05635-f003:**
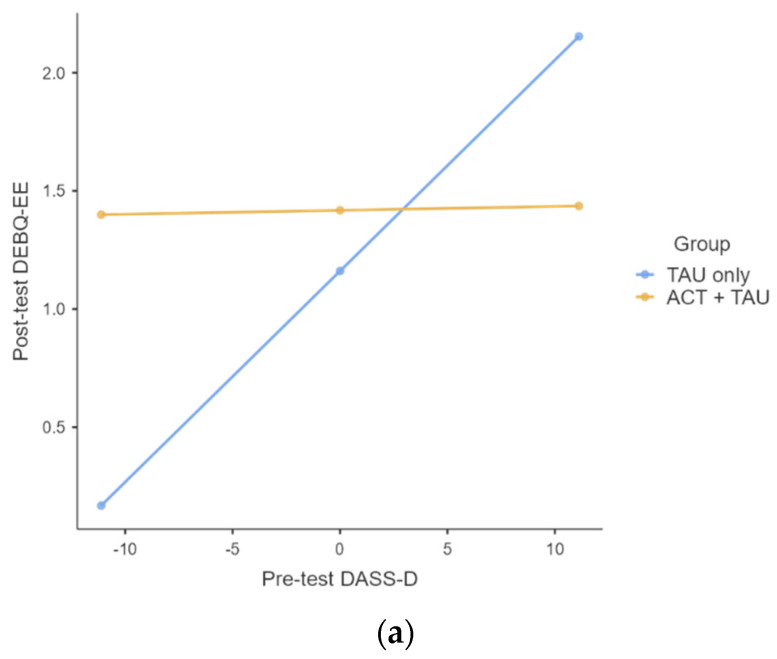
Moderation effects. (**a**) Moderation effect of group in the relationship between depression and emotional eating. Note: DASS-D: Depression subscale of the Depression Anxiety and Stress scale; DEBQ-EE: Emotional Eating subscale of the Dutch Eating Behavior Questionnaire. (**b**) Moderation effect of group in the relationship between stress and emotional eating. Note: DASS-S: Stress subscale of the Depression Anxiety and Stress scale; DEBQ-EE: Emotional Eating subscale of the Dutch Eating Behavior Questionnaire. (**c**) Moderation effect of group in the relationship between experiential avoidance and fusion and emotional eating. Note: AFQ-Y: Avoidance and Fusion Questionnaire for Youth; DEBQ-EE: Emotional Eating subscale of the Dutch Eating Behavior Questionnaire.

**Table 1 ijerph-19-05635-t001:** Pre-test participant characteristics divided by treatment groups (ACT+TAU vs. TAU).

Variables	Group	N	Mean	*SD*	*t*	*p*	Cohen’s d
PWB	TAU	17	46.38	5.19	0.6902	0.495	0.2404
	ACT+TAU	17	44.94	6.61			
DASS-D	TAU	17	14.71	11.40	0.4265	0.673	0.1463
	ACT+TAU	17	13.06	11.12			
DASS-A	TAU	17	14.35	13.18	0.2133	0.832	0.0732
	ACT+TAU	17	13.53	8.93			
DASS-S	TAU	17	18.71	10.15	0.0955	0.924	0.0328
	ACT+TAU	17	18.35	11.36			
DERS	TAU	17	90.24	29.14	−1.1235	0.270	−0.3854
	ACT+TAU	17	101.71	30.37			
AFQ-Y	ACT+TAU	17	29.88	15.94	−0.9238	0.363	−0.3169
	TAU	17	25.00	14.86			
DEBQ-EE	ACT+TAU	17	1.67	1.36	−0.3053	0.762	−0.1047
	TAU	17	1.52	1.38			

Note: PWB: Psychological Well-Being; DASS-D: Depression subscale of the Depression Anxiety and Stress scale; DASS-A: Anxiety subscale of the Depression Anxiety and Stress scale; DASS-S: Stress subscale of the Depression Anxiety and Stress scale; DERS: Difficulties in Emotion Regulation Scale; AFQ-Y: Avoidance and Fusion Questionnaire for Youth; and DEBQ-EE: Emotional Eating subscale of the Dutch Eating Behavior Questionnaire.

**Table 2 ijerph-19-05635-t002:** Differences between ACT+TAU vs TAU only from pre-to-post intervention in all the outcome variables.

Variable	Group	N	Pre-Test		Post-Test	
			Mean	*SD*	Mean	*SD*
PWB	ACT+TAU	17	44.9	6.61	46.7	5.68
	TAU	17	46.4	5.19	46.7	5.93
DASS-D	ACT+TAU	17	13.1	11.1	11.6	13.3
	TAU	17	14.7	11.4	12.5	12.5
DASS-A	ACT+TAU	17	13.5	8.93	40.2	11.1
	TAU	17	14.4	13.2	32.9	8.34
DASS-S	ACT+TAU	17	18.4	11.4	16.0	11.5
	TAU	17	18.7	10.1	15.6	11.0
DERS	ACT+TAU	17	102	30.4	101	32.8
	TAU	17	90.2	29.1	84.8	25.4
AFQ-Y	ACT+TAU	17	29.9	15.9	24.9	17.3
	TAU	17	25.0	14.9	23.6	13.3
DEBQ-EE	ACT+TAU	17	1.67	1.36	1.38	1.30
	TAU	17	1.52	1.38	1.27	1.25

Note: PWB: Psychological Well-Being; DASS-D: Depression subscale of the Depression Anxiety and Stress scale; DASS-A: Anxiety subscale of the Depression Anxiety and Stress scale; DASS-S: Stress subscale of the Depression Anxiety and Stress scale; DERS: Difficulties in Emotion Regulation Scale; AFQ-Y: Avoidance and Fusion Questionnaire for Youth; and DEBQ-EE: Emotional Eating subscale of the Dutch Eating Behavior Questionnaire. Data are reported in means and standard deviations.

## Data Availability

Data collected in this study will be available on request from author A.G.U. with the permission of author A.S. The data will not be publicly available due to privacy/ethical restrictions.

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
