# Peer review of "Preliminary Results from the ACTyourCHANGE in Teens Protocol: A Randomized Controlled Trial Evaluating Acceptance and Commitment Therapy for Adolescents with Obesity"

_ijerph, 2022, doi:10.3390/ijerph19095635_

Round 1

Reviewer 1 Report

I found this manuscript to be dense and hard to understand, not because of the content matter as much as the writing style. Although there are minor grammatical errors, the writing did not flow well. In terms of scientific merit, I believe that the article has some merit, though I feel the authors attempted too many analyses on a small sample size. Overall, I feel the article needs to be improved on to be worthy of publication.  I do feel the manuscript would need extensive revision to make it publishable and I feel the categories provided were restrictive in that sense. I had not given more specific comments highlighting areas related to the strengths and weaknesses of this paper as they are too numerous to mention. Although the study is not without merit, a lot of work would need to go into making it worthy of publication.

Reviewer 2 Report

The authors do a good job of covering a topic that is very important nowadays like obesity. This study was well-designed and analyzed. More importantly, it addresses an important issue: children and teenagers' well-being. The authors are to be commended; this paper is refreshing and insightful. The story is clear, but there are some doubts about the methods and the results.

Treatment-As-Usual (TAU) means the usual treatment, according to accepted standards for your particular discipline, but in this study, it would be better to specify the type of treatment, because this is not clear. The authors talk about a standard multidisciplinary rehabilitation program for weight loss, what exactly does it consist of?

The authors say that the main purpose of this study was to assess preliminary evidence of the efficacy of an ACT-based intervention added to a standard multidisciplinary rehabilitation program for weight loss in a group of adolescents with obesity to improve their psychological conditions but only emotional eating had a decrease and, as the authors themselves say, if it is not directly attributable to our intervention as they could alternatively be attributable to other factors, such as hospitalization, which do not allow patients to have free access to food, even in the case of emotional impulses and other factors. I find this result a bit poor.

Reviewer 3 Report

Authors have done a study to evaluate the efficacy of an Acceptance and Commitment Therapy-based intervention combined with treatment as usual (ACT+TAU) compared to TAU in improving psychological well-being, psychological distress, experiential avoidance and fusion, emotion dysregulation and emotional eating in a sample of 34 in-patient adolescents with obesity at two different periods of time, pre-test and post-test after 3 weeks of analysis. Only a significant effect on anxiety and in the emotional eating were found for ACT+TAU intervention. However, for TAU group higher levels of depression, stress, and experiential avoidance and fusion at pre-test were significantly associated with higher emotional eating at post-test.

I suggest authors to do a minor revision to improve the quality of the manuscript before being accepted.

  1. Change completely the Figure 1, it is very big and the words are out of place and the separations between titles and subtitles are not adequate. The table is difficult to understand, and the meaning of the arrows is not well understood (change it to x as in the rest of the table). Make the box where demographic and physical data appear smaller. Put in parentheses PWB, DASS-21, DERS, AFQ-Y and DEBQ-EE. Rename the footer.
  2. I believe that the study conducted by the authors is not significant enough to draw conclusions that ACT-TAU treatment produces significant results in emotional appetite, since N is very low, only 34 participants, and because a study of just 3 weeks is not enough to see great results. In other words, there is little time (3 weeks) to observe good results in weight and BMI reduction, which improves the participant's self-esteem and can thus improve their psychological and emotional health, thus producing a greater reduction in their emotional appetite.
    Have you planned to conduct this study with more participants and for a longer period as 12 or 24 months?
  3. Saying in the conclusions that this study has produced a significant reduction in emotional appetite with ACT+TAU in a small group of participants and with only 3 weeks of intervention seems very risky to me. In addition, you do not include in the manuscript the weight and BMI data at the beginning and at the end of the treatment, because if the participants have been able to reduce weight and BMI in these 3 months, this resuls could be used to compare and draw conclusions regarding emotional appetite. I suggest incorporating a table of anthropometric results such as height, weight, BMI, tension, etc. of the participants and include in the text a discussion and conclusions of the results obtained and compare it with the results obtained through the questionnaires about the improvement of emotional appetite, stress, anxiety, etc.

Round 2

Reviewer 1 Report

Overall the article reads much better now and is easier to comprehend.